# UniCLIP: Unified Framework for Contrastive Language–Image Pre-training

**Janghyeon Lee**[*]
LG AI Research
janghyeon.lee@lgresearch.ai

**Jongsuk Kim**[*][†]
KAIST
jskpop@kaist.ac.kr

**Hyounguk Shon**[†]
KAIST
hyounguk.shon@kaist.ac.kr

**Bumsoo Kim**
LG AI Research
bumsoo.kim@lgresearch.ai

**Seung Hwan Kim**
LG AI Research
sh.kim@lgresearch.ai

**Honglak Lee**
LG AI Research
honglak@lgresearch.ai

**Junmo Kim**
KAIST
junmo.kim@kaist.ac.kr

## Abstract

Pre-training vision–language models with contrastive objectives has shown promising results that are both scalable to large uncurated datasets and transferable to many downstream applications. Some following works have targeted to improve data efficiency by adding self-supervision terms, but inter-domain (image–text) contrastive loss and intra-domain (image–image) contrastive loss are defined on individual spaces in those works, so many feasible combinations of supervision are overlooked. To overcome this issue, we propose UniCLIP, a Unified framework for Contrastive Language–Image Pre-training. UniCLIP integrates the contrastive loss of both inter-domain pairs and intra-domain pairs into a single universal space. The discrepancies that occur when integrating contrastive loss between different domains are resolved by the three key components of UniCLIP: (1) augmentation-aware feature embedding, (2) MP-NCE loss, and (3) domain dependent similarity measure. UniCLIP outperforms previous vision–language pre-training methods on various single- and multi-modality downstream tasks. In our experiments, we show that each component that comprises UniCLIP contributes well to the final performance.

## 1 Introduction

Recent advances in deep learning have shown significant progress in pre-training large-scale models that transfer well to various downstream applications. Following the success of this paradigm in both fields of computer vision and natural language processing, vision–language pre-training models [9, 20] that learn image representations from natural language supervision have been proposed. In those works, pre-training is done under a simple contrastive loss that makes the embedding of an image and its matching text description (positive pair) more similar to each other than other arbitrary image–text pairs (negative pairs).

Towards a more data-efficient pre-training objective, subsequent works [13, 16] introduced additional self-supervision terms to the image–text contrastive loss, including self-supervision for augmented

---

[*]Equal contribution. Alphabetical order.
[†]Work done during an internship at LG AI Research.

36th Conference on Neural Information Processing Systems (NeurIPS 2022).

images [3, 4], augmented texts [28], and masked texts [13]. Involving more pairs of positive/negative supervisions into the final contrastive loss leads to a more mathematically pleasing objective [3], thus enabling the model to be more data-efficient. Yet, these works entail a major limitation since the contrastive loss for intra-domain pairs, such as image–image pairs, and inter-domain pairs, such as image–text pairs, are defined independently in separated spaces. This means that the contrastive loss is unaware of a substantial set of feasible combinations for negative supervision, for instance image–image pairs are not included when calculating the contrastive loss for image–text supervision, leaving a huge room for improvement in terms of data-efficiency and feature-diversity. Based on this observation, we set the goal of this paper to build a contrastive image–text pre-training framework where the contrastive learning of all possible intra-domain and inter-domain pairs is defined in the same single unified embedding space.

Though this goal sounds intuitive, defining a contrastive loss between the multiple modalities in a unified space has several challenges. First, misalignments can occur between the image–text semantics when applying image augmentations. For example in Figure 1, the semantic of 'a red apple is on the right of sliced green apples' can be easily broken by simple image augmentations like horizontal flipping, converting to grayscale, or cropping, whereas they are fundamental augmentations used in image–image contrastive self-supervised learning [3]. We validate from our experiments that leaving this discrepancy unattended hinders training and degrades final performance. Secondly, existing contrastive losses in literature for multi-positive pairs [10, 15] are not compatible with our training objective that deals with embedding of different modalities. This is because intra-domain pairs, like two different augmented views of a single image, serve as relatively easier examples than inter-domain pairs like image–text pairs. Existing losses [10, 15] are vulnerable to this condition as easy-positive examples and hard-positive examples interfere with each other. Lastly, we discovered that applying the same similarity measure between embeddings from different modalities in our contrastive loss results in a suboptimal performance, because there are

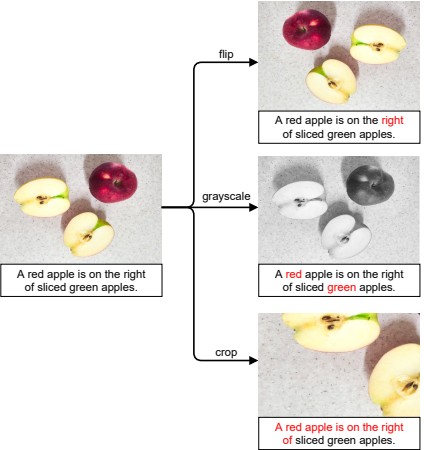

Figure 1: Image–text misalignments caused by data augmentations. The misaligned texts are highlighted in red (best viewed in color).

inherent differences in similarity measures between inter-domain and intra-domain pairs, *i.e.*, samples in an intra-domain pair can be arbitrarily close but samples in an inter-domain pair cannot.

In this paper, we propose UniCLIP: a **Uni**fied framework for **C**ontrastive **L**anguage–**I**mage **P**re-training, that unifies contrastive objectives between multiple modalities on a single embedding space. Each challenge above is addressed with our key components of UniCLIP: (1) **augmentation-aware feature embedding** that makes UniCLIP aware of misalignments caused by data augmentations, (2) **MP-NCE loss** that is designed to stabilize training for both easy- and hard-positive pairs, and (3) **domain dependent similarity measure** that adjusts the difference in similarity scales between inter-domain pairs and intra-domain pairs. UniCLIP outperforms existing vision–language pre-training methods in various single- and multi-modal downstream tasks such as linear probing, zero-shot classification, fine-tuning, and image–text retrieval, by addressing the three problems described above. We validate that each component of UniCLIP successfully addresses the issues of contrastive learning in a unified space and meaningfully contributes to the final performance. Our contribution is summarized as follows:

- We propose UniCLIP, a unified framework for visual–language pre-training that improves data-efficiency by integrating contrastive losses defined across multiple domains into a single universal space. We study new technical challenges that occur from this integration.

- We design new components for UniCLIP to address the aforementioned challenges: augmentation-aware feature embedding, MP-NCE loss, and domain dependent similarity measure. Our extensive experiments show that each of our proposed components serves a key role in the final performance.

- UniCLIP outperforms existing vision–language pre-training methods across multiple downstream tasks that include various modalities.

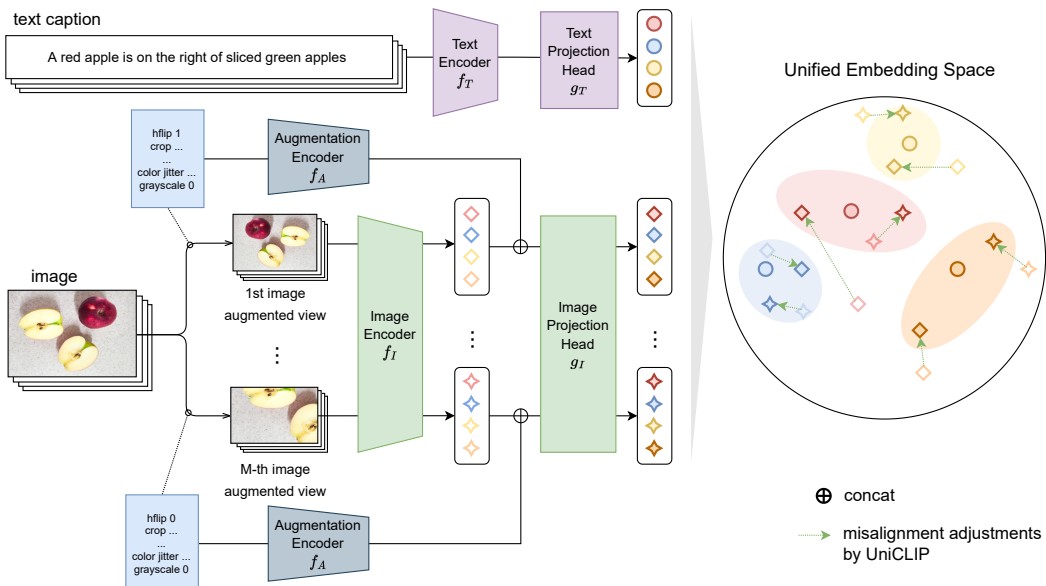

Figure 2: Overview of the UniCLIP framework.

## 2 Methods

The UniCLIP architecture (Figure 2) consists of an augmentation encoder $f_A$, an image encoder $f_I$, a text encoder $f_T$, and corresponding projection heads $g_I$ and $g_T$. $f_I$ encodes an image to an augmentation-agnostic image representation and then $g_I$ outputs an augmentation-aware image embedding. For text caption data, $f_T$ and $g_T$ produce text embeddings on the *same* embedding space as the image embedding space. Image and text representations are learned by our multi-positive NCE loss with domain-dependent similarity scores measured on the unified embedding space. Each element of our method is described in detail in the following sections.

### 2.1 Architecture

**Augmentation Encoder**  To enable an augmentation instruction $\mathcal{A}$ to be used as an input to a network, we first describe it as a real vector containing information about how much each basic transformation in $\mathcal{A}$ is applied to data. For example, image augmentations that frequently appear in contrastive learning can be converted to real vectors as follows:

- *Crop & Resize*: A `RandomResizedCrop` augmentation is encoded to a four-dimensional vector of $(x, y, w, h)$, where $(x, y)$ is the top left corner coordinate of a cropped image and $(w, h)$ is the size of the cropped image, in a normalized coordinate system (*i.e.*, the top left corner of the original image is $(0, 0)$ and the bottom right corner is $(1, 1)$).

- *Color Jitter*: As a `ColorJitter` augmentation changes the brightness, contrast, saturation, and hue of an image, this augmentation is encoded to a four-dimensional vector consisting of the changes in those four factors.

- *Gaussian Blur*: A `GaussianBlur` augmentation is encoded to the standard deviation of its Gaussian blurring kernel.

- *Horizontal Flip*: A `RandomHorizontalFlip` augmentation is encoded to $1$ if an image is actually flipped and $0$ otherwise.

- *Grayscale Convert*: A `RandomGrayscale` augmentation is encoded to $1$ if an image is actually converted to grayscale and $0$ otherwise.

If an image augmentation $\mathcal{A}$ is composed of all five augmentations described above, $\mathcal{A}$ will be first encoded to an 11-dimensional vector according to the above rules and then pass through an MLP to obtain the augmentation embedding $f_A(\mathcal{A})$. Note that $f_A(\mathcal{A})$ will be different for each forward and each sample because of the randomness of the augmentation.

**Image Encoder & Image Projection Head**    For the model to learn how to adjust for image–text misalignment caused by image augmentations, the image encoder or projection head must take the augmentation information as input. However, the encoder cannot fully benefit from augmented data if it knows which augmentation was applied to the image. For example, when the encoder is trained with horizontal flip augmentation, and if it takes an augmented image and an flag of whether the image is flipped or not as input of the form (image, not flipped flag) or (flipped image, flipped flag), then the encoder may exhibit undesirable behavior when it has to encode (flipped image, not flipped flag) from some downstream task, since the encoder was not trained on this kind of data, which means that the model has lost some generalization ability. Therefore, the image encoder must be augmentation-*agnostic* and the image projection head must be augmentation-*aware*. In this way, the encoder can fully enjoy the benefits of data augmentation and generalizes better, while the projection head is still able to correct inter-domain misalignments caused by the augmentations.

To make image representations augmentation-agnostic and image embeddings augmentation-aware, the augmentation information is provided only to the projection head, whereas the encoder only sees the augmented image without knowing which augmentation has been applied. Therefore, for an image $x$, the image encoder $f_I$ takes an augmented image $\mathcal{A}(x)$ as input to get an augmentation-agnostic image representation $h = f_I(\mathcal{A}(x))$. Then, an augmentation-aware image embedding $z = g_I(f_I(\mathcal{A}(x)), f_A(\mathcal{A}))$ in the unified embedding space is obtained from the image representation $h$ and the augmentation embedding $f_A(\mathcal{A})$ by the image projection head $g_I$. We adopt ViT (Vision Transformer) [7] as the image encoder $f_I$ with learnable positional embeddings and the image projection head $g_I$ is composed of three residual blocks. The last activation value of the `[cls]` token is used as the image representation $h$.

**Text Encoder & Text Projection Head**    A raw text is first tokenized by byte pair encoding and wrapped with a start token and an end token, resulting in a tokenized text $x$. Any text augmentation method can also be applied here as in the case of the image embedding, but we do not create multiple augmented views for a text as we found it not very helpful. So, the text representation $h = f_T(x)$ and the text embedding $z = g_T(f_T(x))$ in the unified latent space are obtained without any augmentation embedding. We use Transformer [27] for the text encoder $f_T$ with learnable positional embeddings and a linear layer for the text projection head $g_T$. The last activation value of the start token is used as the text representation $h$.

## 2.2   Contrastive Loss Functions for Multiple Positive Pairs

Contrastive loss functions can be classified according to the number of positive and negative pairs taken by the loss for one data point. For example, triplet loss [22] takes only a single positive pair and a single negative pair, $N$-pair loss [24] and InfoNCE loss [26] take a single positive pair and multiple negative pairs, and MIL-NCE loss [15] and SupCon loss [10] take multiple positive pairs and multiple negative pairs. As there are multiple positive pairs in our unified framework, we first review MIL-NCE loss and SupCon loss functions and discuss their drawbacks.

For an $i$-th embedding $z_i$ in a batch of embeddings $\{z_i\}_i$, let $P_i$ be the set of all positive sample indices of the $i$-th sample excluding $i$ itself and $N_i$ be the set of all negative sample indices of the $i$-th sample.

$$P_i = \{j | (z_i, z_j) \text{ is a positive pair and } j \neq i\} \tag{1}$$
$$N_i = \{j | (z_i, z_j) \text{ is a negative pair}\} \tag{2}$$

A similarity score between the $i$-th and $j$-th embedding is denoted by $s_{i,j} > 0$. A contrastive loss function will try to maximize the similarity scores of positive pairs, while minimize the similarity scores of negative pairs. For example, if there is only one positive sample for each sample in a batch, say $P_i = \{p_i\}$, then InfoNCE loss [26] or NT-Xent loss [3] for the $i$-th sample can be described by

$$\mathcal{L}_i^{\text{InfoNCE}} = -\log \frac{s_{i,p_i}}{s_{i,p_i} + \sum_{n \in N_i} s_{i,n}}. \tag{3}$$

**MIL-NCE Loss**    MIL-NCE loss [15] for the $i$-th embedding is defined by

$$\mathcal{L}_i^{\text{MIL-NCE}} = -\log \frac{\sum_{p \in P_i} s_{i,p}}{\sum_{p \in P_i} s_{i,p} + \sum_{n \in N_i} s_{i,n}}. \tag{4}$$

The MIL-NCE loss function is configured to maximize the sum of all positive pair similarity scores $\sum_{p \in P_i} s_{i,p}$ and minimize the sum of all negative pair similarity scores $\sum_{n \in N_i} s_{i,n}$. However, hard positive pairs cannot receive enough gradients from $\mathcal{L}_i^{\text{MIL-NCE}}$ when there are easy positive pairs whose similarity scores are sufficiently large to dominate the numerator and denominator, as the MIL-NCE loss compares negative pairs with the sum of positive scores $\sum_{p \in P_i} s_{i,p}$ only, not each positive pair $s_{i,p}$ individually. For some $q \in P_i$, the gradient from $\mathcal{L}_i^{\text{MIL-NCE}}$ to $s_{i,q}$ is

$$\frac{\partial \mathcal{L}_i^{\text{MIL-NCE}}}{\partial s_{i,q}} = -\frac{\sum_{n \in N_i} s_{i,n}}{\left(\sum_{p \in P_i} s_{i,p}\right)\left(\sum_{p \in P_i} s_{i,p} + \sum_{n \in N_i} s_{i,n}\right)}, \tag{5}$$

therefore the gradient will vanish to zero when $\sum_{p \in P_i} s_{i,p}$ is already large because of easy positive pairs even if the positive pair's score $s_{i,q}$ is small. In other words, *easy positive pairs hinder the training of hard positive pairs* in MIL-NCE loss. This problem will be more pronounced in our unified framework because hard positives and easy positives frequently coexist with supervisions from intra-domain and inter-domain.

**SupCon Loss**  SupCon loss [10] for the $i$-th embedding is described by

$$\mathcal{L}_i^{\text{SupCon}} = \mathbb{E}_{p \in P_i}\left[-\log \frac{s_{i,p}}{\sum_{p' \in P_i} s_{i,p'} + \sum_{n \in N_i} s_{i,n}}\right]. \tag{6}$$

In this case, each positive score $s_{i,p}$ is compared with the negative pairs, but the sum of the positive scores in the denominator still causes an undesirable side effect. For an easy positive pair with a large similarity score, it can be possible to decrease the loss by decreasing its score and so the denominator. For $q \in P_i$,

$$\frac{\partial \mathcal{L}_i^{\text{SupCon}}}{\partial s_{i,q}} = \frac{s_{i,q} - \frac{1}{|P_i|}\left(\sum_{p \in P_i} s_{i,p} + \sum_{n \in N_i} s_{i,n}\right)}{s_{i,q}\left(\sum_{p \in P_i} s_{i,p} + \sum_{n \in N_i} s_{i,n}\right)}, \tag{7}$$

so hard positives would be trained better than MIL-NCE loss because of a relatively large update by the $s_{i,q}$ term in the denominator. However, if we assume the sum of positive scores is much greater than the sum of negative scores, then

$$\frac{\partial \mathcal{L}_i^{\text{SupCon}}}{\partial s_{i,q}} \propto s_{i,q} - \frac{1}{|P_i|}\left(\sum_{p \in P_i} s_{i,p} + \sum_{n \in N_i} s_{i,n}\right) \approx s_{i,q} - \mathbb{E}_{p \in P_i}\left[s_{i,p}\right]. \tag{8}$$

As gradient is not always negative, $\mathcal{L}_i^{\text{SupCon}}$ will try to decrease the similarity score of an easy positive pair $(z_i, z_q)$ since $s_{i,q}$ will be larger than the average positive score, instead of increasing or at least maintaining it. In other words, *hard positive pairs hinder the convergence of easy positive scores* in SupCon loss.

**Multi-positive NCE Loss**  As the sum of the positive scores in the denominator causes easy and hard positive pairs to interfere with each other, we can just use a multi-positive version of InfoNCE loss to make each positive pair independently contribute to the loss as follows.

$$\mathcal{L}_i = \mathbb{E}_{p \in P_i}\left[-\log \frac{s_{i,p}}{s_{i,p} + \sum_{n \in N_i} s_{i,n}}\right] \tag{9}$$

As can be seen in the gradient

$$\frac{\partial \mathcal{L}_i}{\partial s_{i,q}} = -\frac{\sum_{n \in N_i} s_{i,n}}{|P_i| s_{i,q}\left(s_{i,q} + \sum_{n \in N_i} s_{i,n}\right)}, \tag{10}$$

hard positive samples can be trained with sufficiently large update from the $s_{i,q}$ term in the denominator, and the decreasing easy positive pair similarity problem does not occur as the gradient is always negative.

With this multi-positive version of InfoNCE loss, we reconsider excluding $i$ from the positive set $P_i$ in Equation 1. If a contrastive loss can handle multiple positive pairs, then there is no reason to

exclude the trivial pair $(z_i, z_i)$ from the loss definition. Since $z_i$ is most similar to $z_i$ itself, the trivial pair must be also utilized as a strong positive pair, which will result in

$$\mathcal{L}_i = \mathbb{E}_{p \in P_i \cup \{i\}} \left[ -\log \frac{s_{i,p}}{s_{i,p} + \sum_{n \in N_i} s_{i,n}} \right]. \tag{11}$$

Here, we propose a multi-positive NCE loss for our unified contrastive learning framework called MP-NCE loss, which is a weighted version of Equation 11 defined as

$$\mathcal{L}_i^{\text{MP-NCE}} = \mathbb{E}_{p \in P_i \cup \{i\}} \left[ -w_{\mathcal{D}(i,p)} \log \frac{s_{i,p}}{s_{i,p} + \sum_{n \in N_i} s_{i,n}} \right], \tag{12}$$

where $\mathcal{D}(i, p)$ indicates the domain combination from which the $i$-th and $p$-th data were sampled, and $w_{\mathcal{D}(i,p)}$ is a domain-specific balancing hyperparameter which makes each inter-domain and intra-domain supervision equally contributes to the loss. For example, when we use three augmented views of an image and one corresponding text for each original image–text pair from dataset, there are a total of $9N$ image–image positive pairs, $6N$ image–text positive pairs, and $N$ text–text positive pairs in a batch, so $w_{\mathcal{D}(i,p)}$ is set to $1/9$, $1/6$, $1$ if $(z_i, z_p)$ is an image–image pair, image–text pair, text–text pair, respectively.

Although we have proposed MP-NCE loss in a multi-positive setting, one should consider using MP-NCE loss even in single positive settings, such as image self-supervised contrastive learning, by treating a trivial pair $(z_i, z_i)$ as positive as well since MP-NCE involves negligible computational overhead compared to backbone networks.

## 2.3 Domain-Dependent Similarity Score

In SimCLR [3] and CLIP [20], the similarity score $s_{i,j}$ between the $i$-th embedding $z_i$ and $j$-th embedding $z_j$ is defined by

$$s_{i,j} = \exp\left( \frac{1}{\tau} \cdot \frac{z_i^\top z_j}{\|z_i\|\|z_j\|} \right), \tag{13}$$

where $\tau$ is a positive real number, usually smaller than 1. As the cosine similarity of two embeddings cannot have a value outside the interval $[-1, 1]$, the cosine similarity is divided by the temperature $\tau$ to extend its range. $\tau$ can be a pre-defined hyperparameter, or can rather be a learnable parameter allowing the model to choose an appropriate scale for the convergence of a contrastive loss.

To classify an input pair $(z_i, z_j)$ as positive or negative, we can define a threshold $b$ and classify it as positive if the cosine similarity between $z_i$ and $z_j$ is greater than $b$, and negative otherwise. We may absorb this threshold $b$ into the similarity score as an offset like

$$s_{i,j} = \exp\left( \frac{1}{\tau} \left( \frac{z_i^\top z_j}{\|z_i\|\|z_j\|} - b \right) \right), \tag{14}$$

and expect that the optimal threshold will be learned by the model, as in the case of the temperature. Note that the temperature will amplify the score if the cosine similarity is greater than $b$ otherwise reduce it, so Equation 14 is a reasonable similarity measure with which the threshold can be treated as a decision boundary for the binary classification problem. However, unfortunately, the offset $b$ does not contribute to InfoNCE loss (Equation 3) at all since $b$'s in the numerator and denominator cancel out as

$$\mathcal{L}_i^{\text{InfoNCE}} = -\log \frac{s_{i,p_i}}{s_{i,p_i} + \sum_{n \in N_i} s_{i,n}} = -\log \frac{\exp(b/\tau) s_{i,p_i}}{\exp(b/\tau) s_{i,p_i} + \sum_{n \in N_i} \exp(b/\tau) s_{i,n}} \tag{15}$$

for any $\tau$ and $b$, which means $\partial \mathcal{L}_i^{\text{InfoNCE}} / \partial b$ is always zero.

On the other hand, when data pairs are sampled from multiple domains as in our unified framework, the threshold can be different depending on whether the sampled data pair is an intra-domain pair or an inter-domain pair, as it would be easier to classify intra-domain positive pairs than inter-domain positive pairs in general. This motivates us to introduce domain-specific temperature $\tau_{\mathcal{D}(i,j)}$ and offset $b_{\mathcal{D}(i,j)}$, and propose a domain-dependent similarity score

$$s_{i,j} = \exp\left( \frac{1}{\tau_{\mathcal{D}(i,j)}} \left( \frac{z_i^\top z_j}{\|z_i\|\|z_j\|} - b_{\mathcal{D}(i,j)} \right) \right). \tag{16}$$

Table 1: Zero-shot image classification performance and linear probing performance on 11 down-stream datasets. [†]Results reported in the original paper.

| Method | Pre-train dataset | Pets | CIFAR-10 | CIFAR-100 | SUN397 | Food-101 | Flowers | Cars | Caltech-101 | Aircraft | DTD | ImageNet | Average |
|---|---|---|---|---|---|---|---|---|---|---|---|---|---|
| *Zero-shot classification:* | | | | | | | | | | | | | |
| CLIP-ViT-B/32 | YFCC15M | 19.4 | 62.3 | 33.6 | 40.2 | 33.7 | 6.3 | 2.1 | 55.4 | 1.4 | 16.9 | 31.3 | 27.5 |
| SLIP-ViT-B/32 | YFCC15M | 28.3 | 72.2 | 45.3 | 45.1 | 44.7 | 6.8 | 2.9 | 65.9 | 1.9 | 21.8 | 38.3 | 33.9 |
| DeCLIP-ViT-B/32 | YFCC15M | 30.2 | 72.1 | 39.7 | **51.6** | 46.9 | 7.1 | **3.9** | 70.1 | 2.5 | **24.2** | 41.2 | 35.4 |
| UniCLIP-ViT-B/32 | YFCC15M | **32.5** | **78.6** | **47.2** | 50.4 | **48.7** | **8.1** | 3.4 | **73.0** | **2.8** | 23.3 | **42.8** | **37.3** |
| DeCLIP-ResNet50[†] [13] | Open30M | - | - | - | - | - | - | - | - | - | - | 49.3 | - |
| UniCLIP-ViT-B/32 | Open30M | 69.2 | 87.8 | 56.5 | 61.1 | 64.6 | 8.0 | 19.5 | 84.0 | 4.7 | 36.6 | **54.2** | 49.7 |
| *Linear probing:* | | | | | | | | | | | | | |
| CLIP-ViT-B/32 | YFCC15M | 71.2 | 89.2 | 72.1 | 70.1 | 71.4 | 93.2 | 34.9 | 84.3 | 29.7 | 60.9 | 61.1 | 67.1 |
| SLIP-ViT-B/32 | YFCC15M | 75.4 | 90.5 | 75.3 | 73.5 | 77.1 | 96.1 | 43.0 | 87.2 | 34.1 | 71.1 | 68.1 | 71.9 |
| DeCLIP-ViT-B/32 | YFCC15M | 76.5 | 88.6 | 71.6 | 75.9 | 79.3 | 96.7 | 42.6 | 88.0 | 32.6 | 69.1 | 69.2 | 71.8 |
| UniCLIP-ViT-B/32 | YFCC15M | **83.1** | **92.5** | **78.2** | **77.0** | **81.3** | **97.1** | **49.8** | **88.9** | **36.2** | **72.8** | **70.8** | **75.2** |
| UniCLIP-ViT-B/32 | Open30M | 85.4 | 95.1 | 81.5 | 79.2 | 84.4 | 97.3 | 67.3 | 91.1 | 39.0 | 77.2 | 74.0 | 79.1 |

For image–text unified contrastive learning, we have three possible domain combinations, so there will be three different temperatures and three offsets respectively for image–image pairs, image–text pairs, and text–text pairs.

With the proposed domain-dependent similarity score (Equation 16) and MP-NCE loss (Equation 12), the offsets are no longer cancelled out as negative pairs are sampled from multiple different domains. Specifically, because any real number can be added to the cosine similarity term as in Equation 15 without changing the loss function, the offsets lose only 1 intrinsic dimension and thus the model is able to learn *relative* thresholds. In other words, it is now possible to learn the domain-specific offsets so that we can expect the offset of an easier domain combination to be greater than that of harder one.

# 3 Experiments

**Datasets** For reproducibility, we use publicly available datasets for training and evaluation in our experiments, including CC3M [23], CC12M [2], DeCLIP YFCC15M [13, 25] for training and Pets [18], CIFAR-10, CIFAR-100 [12], SUN397 [29], Food-101 [1], Flowers [17], Cars [11], Caltech-101 [8], Aircraft [14], DTD [6], ImageNet-1k [21], Flickr30k [19], COCO Captions [5] for evaluation. We define the union of CC3M, CC12M, and YFCC15M as Open30M dataset.

**Settings** For each original image and corresponding text caption, one weakly augmented image, two strongly augmented images, and one text form a positive group in our experiments. Detailed augmentation and optimization configurations can be found in Appendix.

## 3.1 Main Results

We evaluate the transferability of our model in single-modal and multi-modal downstream tasks. Linear probing and fine-tuning on image classification tasks are performed for single-modal benchmarks, and image–text retrieval tasks and zero-shot image classification tasks are evaluated for multi-modal benchmarks.

**Linear Probing & Fine-Tuning** For single-modal experiments, we remove the image projection head $g_I$ and augmentation encoder $f_A$, and use only the image encoder $f_I$. Table 1 reports linear classification performances on 11 downstream datasets. We report ImageNet fine-tuning accuracy in Table 2. UniCLIP consistently outperforms other methods on all downstream datasets in the single-modal experiments.

Table 2: ImageNet-1k fine-tuning accuracy for the models pre-trained on YFCC15M.

| Method | Accuracy |
|---|---|
| CLIP-ViT-B/32 | 72.27 |
| SLIP-ViT-B/32 | 75.64 |
| DeCLIP-ViT-B/32 | 74.34 |
| UniCLIP-ViT-B/32 | **76.54** |

Table 3: Zero-shot image–text retrieval on the test splits of Flickr30k and COCO Captions with models pre-trained on YFCC15M. [†]Pre-trained on Open30M.

| | Image-to-text retrieval | | | | | | Text-to-image retrieval | | | | | |
| | Flickr30k | | | COCO Captions | | | Flickr30k | | | COCO Captions | | |
| Method | R@1 | R@5 | R@10 | R@1 | R@5 | R@10 | R@1 | R@5 | R@10 | R@1 | R@5 | R@10 |
|---|---|---|---|---|---|---|---|---|---|---|---|---|
| CLIP-ViT-B/32 | 34.9 | 63.9 | 75.9 | 20.8 | 43.9 | 55.7 | 23.4 | 47.2 | 58.9 | 13.0 | 31.7 | 42.7 |
| SLIP-ViT-B/32 | 47.8 | 76.5 | 85.9 | 27.7 | 52.6 | 63.9 | 32.3 | 58.7 | 68.8 | 18.2 | 39.2 | 51.0 |
| DeCLIP-ViT-B/32 | 51.4 | 80.2 | 88.9 | 28.3 | 53.2 | 64.5 | 34.3 | 60.3 | 70.7 | 18.4 | 39.6 | 51.4 |
| UniCLIP-ViT-B/32 | **52.3** | **81.6** | **89.0** | **32.0** | **57.7** | **69.2** | **34.8** | **62.0** | **72.0** | **20.2** | **43.2** | **54.4** |
| UniCLIP-ViT-B/32[†] | 75.6 | 94.2 | 97.3 | 46.1 | 74.0 | 83.0 | 61.4 | 85.2 | 91.5 | 35.2 | 61.3 | 71.7 |

**Zero-Shot Classification & Image–Text Retrieval**  Table 1 shows zero-shot classification performances on 11 downstream datasets. We perform prompt ensembling for each class with the same prompt templates as [16, 20]. Table 3 shows the results of zero-shot image–text retrieval on Flickr30k and COCO Captions benchmarks.

## 3.2  Ablation Studies

In this section, we report experiments to inspect how each component of UniCLIP contributes to the final performance. We pre-train all variants of UniCLIP with a ViT-B/16 backbone on the CC3M dataset for 50 epochs, and compare their ImageNet-1k zero-shot evaluation performances.

**Image Projection Head Types**  We tried several different architectures for the image projection head including a linear layer, MLP layers, and residual blocks, when the head takes augmentation embeddings as input or not, as in Table 4a. It turns out that MLP shows a strong tendency to overfit, even performs worse than a linear layer. By adding skip connections to the head, it can fully utilize augmentation information with increased capabilities while avoiding overfitting. Making the image projection head augmentation-aware improves the performance for all head types as the head can handle the inter-domain misalignments.

**Augmentation Configurations**  Table 4b studies the effect of augmentation encoding on various augmentation configurations. Using only strongly augmented images severely degrades the performance without augmentation embedding, because strong augmentations will generate more image–text misalignments. Since it is observed that including one weakly augmented image works better than using only strongly augmented ones, we choose to keep one weakly augmented image in the positive set as a stable reference sample.

Table 4: ImageNet-1k zero-shot accuracy with varying image projection head types and augmentation configurations.

(a) **Image projection head types.** One weak and two strong image augmentations are used.

| Augmentation embedding | Head type | Accuracy |
|---|---|---|
| ✗ | MLP 3 layers | 24.01 |
| | MLP 6 layers | 23.62 |
| | 1 ResBlock | **24.76** |
| | 3 ResBlocks | 24.46 |
| ✓ | Linear layer | 24.68 |
| | MLP 3 layers | 24.54 |
| | MLP 6 layers | 24.15 |
| | 1 ResBlock | 27.67 |
| | 3 ResBlocks | **27.84** |

(b) **Augmentation configurations.** 1-ResBlock head is used for no augmentation embedding config and 3-ResBlock head is used with augmentation embedding.

| Augmentation embedding | Augmentation | Accuracy |
|---|---|---|
| ✗ | 3 weak | 24.49 |
| | 1 weak, 2 strong | **24.76** |
| | 3 strong | 22.60 |
| ✓ | 3 weak | 23.40 |
| | 1 weak, 2 strong | **27.84** |
| | 3 strong | 26.43 |

**Domain-dependent Similarity Score and Unified Supervision**    In Table 5, we can see how the performance changes depending on whether the shared similarity score (Equation 14) or the domain-dependent score (Equation 16) is used. We also run experiments where positive and negative sets are formed separately with respect to the domain combination as in SLIP [16] and DeCLIP [13]. The best performance comes out from the domain-dependent similarity measure with unified supervisions, as expected.

Table 5: ImageNet-1k zero-shot accuracy with domain-dependency of similarity score and supervision.

| Temperature and offset | Supervision | Accuracy |
|---|---|---|
| Shared across domains | Unified | 25.51 |
| Domain-dependent | Separated | 26.59 |
| Domain-dependent | Unified | **27.84** |

**Loss Functions**    As analyzed in Section 2.2, SupCon loss [10] outperforms MIL-NCE loss [15], but performs worse than the multi-positive version of InfoNCE loss (Equation 9), as in Table 6. The balancing weight $w_{\mathcal{D}(i,p)}$ can boost the performance, and surprisingly, we can significantly improve performance with negligible additional computations by simply adding a trivial pair $(z_i, z_i)$ to the positive set $P_i$.

Table 6: ImageNet-1k zero-shot accuracy with different loss functions.

| Loss function | Accuracy |
|---|---|
| MIL-NCE | 22.23 |
| SupCon | 23.04 |
| MP-NCE w/o trivial pair $(z_i, z_i)$ and $w_{\mathcal{D}(i,p)}$ (Eq. 9) | 24.60 |
| MP-NCE w/o $w_{\mathcal{D}(i,p)}$ (Eq. 11) | 26.41 |
| MP-NCE | **27.84** |

## 4   Conclusion

We have proposed UniCLIP, a unified framework for visual–language pre-training that improves data-efficiency by integrating contrastive losses defined across multiple domains into a single universal space. In this paper, image–text datasets were used to validate our method since vision and language are among the most actively studied fields in deep learning. Although we have experimented with vision–language multimodal datasets only, the proposed UniCLIP framework can be easily extended to other types of multimodal datasets because it is designed in a modality-agnostic way except for the augmentation encoding part. All modality-specific knowledge required to apply UniCLIP to different types of modality is to describe each modality-specific augmentation as a real vector, as in Section 2.1, which is quite straightforward. We leave it for future work to see how well the UniCLIP framework works with various types of multimodal datasets.

## Acknowledgements

This work was supported by Institute of Information & communications Technology Planning & Evaluation (IITP) grant funded by the Korea government(MSIT). (No. 2022-0-00184, Development and Study of AI Technologies to Inexpensively Conform to Evolving Policy on Ethics)

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
