# A  Related Works

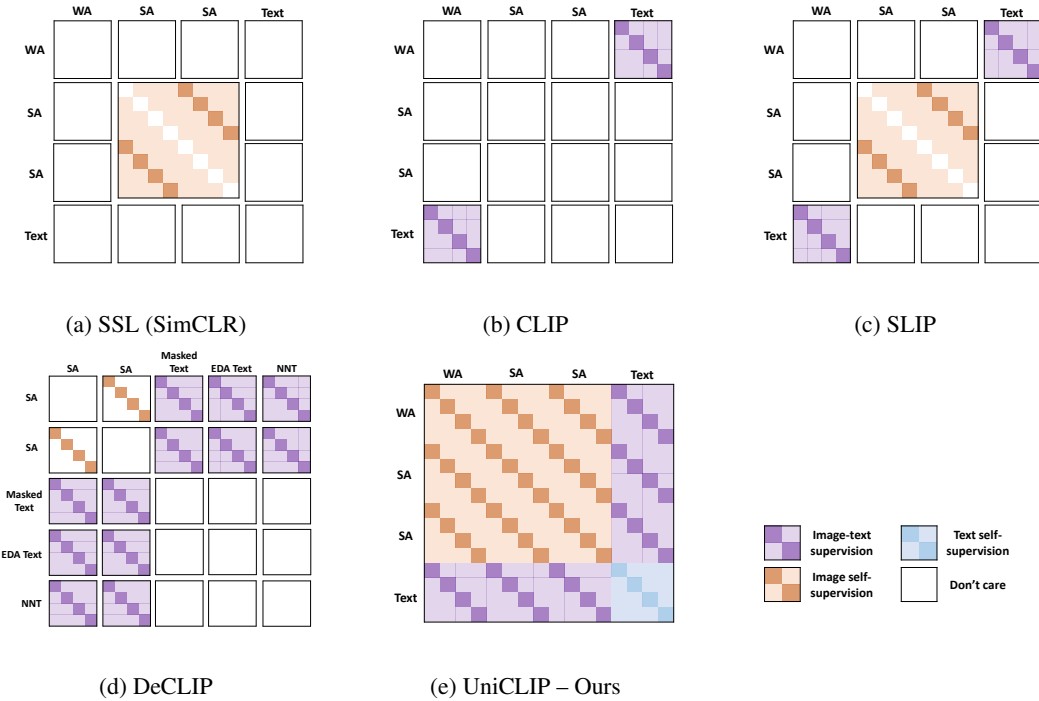

Figure A: Similarity matrices in various contrastive learning methods. Darker colors represent positive pairs and lighter colors represent negative pairs. **WA**: Weakly Augmented image, **SA**: Strongly Augmented image, **NNT**: Nearest Neighborhood Text.

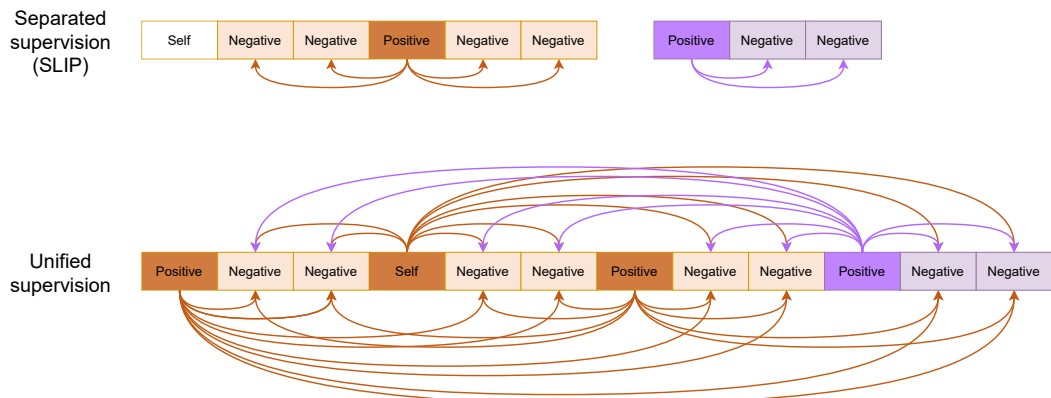

Figure B: Separated supervision and unified supervision. All possible pairs including intra-domain and inter-domain pairs contribute to contrastive learning across different supervisions in our unified framework, whereas each supervision is considered independently in previous works.

**Self-Supervised Learning**    Recently, self-supervised learning (SSL) has drawn a huge attention as a pre-training method that is scalable to large and uncurated datasets. Among the various pretext tasks proposed for self-supervised learning [10, 26, 46, 30, 47, 27, 9, 13], it has been demonstrated that minimizing the contrastive cross entropy loss between positive pairs (*i.e.*, an augmented view of the original data) against negative pairs (*i.e.*, other data samples) yields representations that show solid performance throughout multiple tasks and datasets.

**Contrastive Language–Image Pre-training**    CLIP [32] introduced a new paradigm of pre-training by defining a contrastive loss with large-scale image–text pairs (Figure Ab). Here, the image and its matching text description comprise a positive pair, and the representations of the two are learned to be similar to each other than other arbitrary image–text pairs. CLIP learns powerful representations that are transferable throughout a wide set of datasets and tasks, showing robust performance even in zero-shot evaluations. The image–text representation obtained by CLIP revolutionized multiple research directions in various fields since it provides a standard measure for how semantically similar a given image–text pair is [18, 33].

**Self-Supervised Learning Meets CLIP**    Following works of SLIP [24] and DeCLIP [21] improved CLIP by introducing additional SSL terms to the original image–text contrastive loss formula. However, these methods have limited supervisions since the contrastive loss between inter-domain pairs and intra-domain pairs are defined in separate spaces. To address this issue, our UniCLIP defines the contrastive loss of both inter-domain pairs and intra-domain pairs in a single unified space, utilizing the supervision from all possible combinations throughout multiple domains at once.

# B    Algorithm

The main algorithm of UniCLIP is summarized in Algorithm A.

---

**Algorithm A** UniCLIP

---

**Input:** image encoder $f_I$, text encoder $f_T$, image projection head $g_I$, text projection head $g_T$,
   augmentation encoder $f_A$, batch size $N$, temperature $\tau \in \mathbb{R}^3$, offset $b \in \mathbb{R}^3$,
   weak augmentation distribution $p_{wa}$, strong augmentation distribution $p_{sa}$

1: **for** sampled mini-batch $\{(x_k^I, x_k^T)\}_{k=1}^N$ **do**
2:    **for all** $k \in \{1, \dots, N\}$ **do**
3:       draw augmentation instructions $\mathcal{A}_1 \sim p_{wa}, \mathcal{A}_2 \sim p_{sa}, \mathcal{A}_3 \sim p_{sa}$
4:       $z_k = g_I(f_I(\mathcal{A}_1(x_k^I)), f_A(\mathcal{A}_1))$
5:       $z_{k+N} = g_I(f_I(\mathcal{A}_2(x_k^I)), f_A(\mathcal{A}_2))$
6:       $z_{k+2N} = g_I(f_I(\mathcal{A}_3(x_k^I)), f_A(\mathcal{A}_3))$
7:       $z_{k+3N} = g_T(f_T(x_k^T))$
8:    **end for**
9:    **for all** $i \in \{1, \dots, 4N\}$ **do**
10:      **for all** $j \in \{1, \dots, 4N\}$ **do**

11: 
$$\mathcal{D}(i,j) = \begin{cases} 1, & \text{if } i \le 3N \text{ and } j \le 3N \\ 3, & \text{if } i > 3N \text{ and } j > 3N \\ 2, & \text{otherwise} \end{cases}$$

12: 
$$s_{i,j} = \exp\left( \frac{1}{\tau_{\mathcal{D}(i,j)}} \left( \frac{z_i^\top z_j}{\|z_i\|\|z_j\|} - b_{\mathcal{D}(i,j)} \right) \right)$$

13:      **end for**
14:      $P_i = \{j \in \{1, \dots, 4N\} \setminus \{i\} | (j-i)/N \in \mathbb{Z}\}$
15:      $N_i = \{1, \dots, 4N\} \setminus P_i \setminus \{i\}$
16:      $w = (1/9, 1/6, 1)$
17:      $\mathcal{L}_i = \mathbb{E}_{p \in P_i \cup \{i\}} \left[ -w_{\mathcal{D}(i,p)} \log \frac{s_{i,p}}{s_{i,p} + \sum_{n \in N_i} s_{i,n}} \right]$
18:    **end for**
19:    $\mathcal{L} = \frac{1}{4N} \sum_{i=1}^{4N} \mathcal{L}_i$
20:    update networks, temperature, offset to minimize $\mathcal{L}$
21: **end for**

---

# C  Additional Experimental Results

**Zero-Shot Classification on ImageNet Variations**  We report zero-shot classification performance on ImageNet variations such as ImageNet-R [14], ImageNet-Sketch [43], ImageNetV2 [34], and ImageNet-A [15] in Table A.

Table A: Zero-shot accuracy on ImageNet variations.

| Method | Pre-train dataset | ImageNet | ImageNet-R | ImageNet-Sketch | ImageNetV2 | ImageNet-A |
|---|---|---|---|---|---|---|
| CLIP-ViT-B32 | YFCC15M | 31.3 | 22.6 | 7.2 | 25.5/30.6/33.6 | 8.1 |
| SLIP-ViT-B32 | YFCC15M | 38.3 | 31.7 | 11.9 | 33.2/37.8/41.8 | 13.2 |
| DeCLIP-ViT-B32 | YFCC15M | 41.2 | 34.3 | 14.5 | 35.4/40.4/43.8 | **15.0** |
| UniCLIP-ViT-B32 | YFCC15M | **42.8** | **37.8** | **15.7** | **36.5/41.9/46.3** | 14.4 |
| UniCLIP-ViT-B/32 | Open30M | 54.2 | 61.8 | 36.0 | 47.1/54.0/58.6 | 18.3 |

**Zero-Shot Image–Text Retrieval on Validation Splits**  Table B shows the results of zero-shot image–text retrieval on the validation splits of Flickr30k and COCO Captions benchmarks.

Table B: Zero-shot image–text retrieval on the validation splits of Flickr30k and COCO Captions with models pre-trained on YFCC15M. $^{\dagger}$Pre-trained on Open30M.

| Method | Image-to-text retrieval | | | | | | Text-to-image retrieval | | | | | |
|---|---|---|---|---|---|---|---|---|---|---|---|---|
| | Flickr30k | | | COCO Captions | | | Flickr30k | | | COCO Captions | | |
| | R@1 | R@5 | R@10 | R@1 | R@5 | R@10 | R@1 | R@5 | R@10 | R@1 | R@5 | R@10 |
| CLIP-ViT-B/32 | 37.3 | 66.2 | 77.1 | 20.1 | 42.9 | 55.1 | 24.9 | 49.0 | 60.0 | 13.3 | 31.7 | 42.3 |
| SLIP-ViT-B/32 | 48.7 | 75.2 | 84.7 | 26.9 | 51.9 | 63.8 | 33.1 | 59.0 | 68.8 | 18.2 | 39.6 | 51.1 |
| DeCLIP-ViT-B/32 | 51.3 | 79.3 | 88.7 | 28.1 | 53.6 | 65.2 | 34.8 | 62.2 | 71.5 | 17.9 | 39.8 | 51.6 |
| UniCLIP-ViT-B/32 | **55.7** | **82.9** | **90.0** | **32.0** | **58.8** | **70.3** | **36.7** | **62.6** | **72.4** | **20.3** | **43.1** | **54.5** |
| UniCLIP-ViT-B/32$^{\dagger}$ | 76.7 | 94.2 | 96.9 | 47.8 | 74.4 | 84.2 | 62.4 | 86.7 | 92.2 | 35.4 | 61.6 | 72.0 |

**Fine-tuning Results on Image–Text Retrieval**  We fine-tuned YFCC15M pre-trained models on Flickr30k and COCO Captions for 10 epochs and report the results in Table C and Table D. Our method consistently outperforms on fine-tuned image–text retrieval benchmarks.

Table C: Fine-tuned image–text retrieval on the test splits of Flickr30k and COCO Captions with models pre-trained on YFCC15M. $^{\dagger}$Pre-trained on Open30M.

| Method | Image-to-text retrieval | | | | | | Text-to-image retrieval | | | | | |
|---|---|---|---|---|---|---|---|---|---|---|---|---|
| | Flickr30k | | | COCO Captions | | | Flickr30k | | | COCO Captions | | |
| | R@1 | R@5 | R@10 | R@1 | R@5 | R@10 | R@1 | R@5 | R@10 | R@1 | R@5 | R@10 |
| CLIP-ViT-B/32 | 57.4 | 84.7 | 90.2 | 34.4 | 63.5 | 75.2 | 40.4 | 69.5 | 79.6 | 24.0 | 50.8 | 63.5 |
| SLIP-ViT-B/32 | 68.9 | 91.9 | 95.1 | 43.7 | 71.8 | 82.4 | 51.0 | 79.5 | 86.8 | 31.0 | 58.8 | 70.3 |
| DeCLIP-ViT-B/32 | 73.6 | 93.9 | 97.2 | 47.9 | 75.5 | 84.6 | 55.9 | 83.4 | 90.2 | 33.8 | 62.7 | 74.4 |
| UniCLIP-ViT-B/32 | **77.9** | **95.1** | **98.0** | **52.7** | **78.6** | **87.4** | **61.0** | **85.9** | **92.2** | **37.6** | **66.3** | **77.0** |
| UniCLIP-ViT-B/32$^{\dagger}$ | 87.8 | 98.2 | 99.2 | 62.2 | 85.3 | 91.9 | 70.7 | 91.5 | 95.4 | 45.6 | 73.5 | 82.5 |

Table D: Fine-tuned image–text retrieval on the validation splits of Flickr30k and COCO Captions with models pre-trained on YFCC15M. $^{\dagger}$Pre-trained on Open30M.

| Method | Image-to-text retrieval | | | | | | Text-to-image retrieval | | | | | |
|---|---|---|---|---|---|---|---|---|---|---|---|---|
| | Flickr30k | | | COCO Captions | | | Flickr30k | | | COCO Captions | | |
| | R@1 | R@5 | R@10 | R@1 | R@5 | R@10 | R@1 | R@5 | R@10 | R@1 | R@5 | R@10 |
| CLIP-ViT-B/32 | 58.3 | 84.8 | 91.5 | 36.1 | 65.0 | 76.4 | 43.1 | 71.1 | 80.3 | 24.9 | 51.7 | 64.1 |
| SLIP-ViT-B/32 | 69.6 | 90.4 | 95.7 | 45.0 | 74.0 | 83.0 | 52.1 | 79.4 | 86.9 | 31.6 | 59.5 | 71.3 |
| DeCLIP-ViT-B/32 | 75.6 | 93.0 | 96.6 | 48.7 | 77.3 | 86.2 | 57.8 | 83.3 | 90.3 | 34.2 | 63.1 | 74.6 |
| UniCLIP-ViT-B/32 | **78.1** | **94.9** | **97.7** | **54.5** | **80.9** | **89.1** | **61.0** | **86.0** | **91.9** | **38.0** | **67.2** | **78.0** |
| UniCLIP-ViT-B/32$^{\dagger}$ | 88.0 | 97.7 | 99.3 | 63.0 | 86.3 | 92.4 | 72.1 | 92.1 | 95.9 | 46.2 | 73.7 | 83.1 |

# D  Additional Ablation Studies

**Augmentation-Agnostic vs. Augmentation-Aware Image Encoder**   Making the image encoder augmentation-agnostic is a key design idea for better generalization in a unified contrastive learning framework. To verify this, we feed the image encoder with the augmentation embedding instead of the projection head and performed the same experiment with this augmentation-aware image encoder. As seen in Table E, the augmentation-aware encoder performs much worse than the augmentation-agnostic encoder.

Table E: ImageNet-1k zero-shot accuracy with respect to augmentation-awareness of image encoder.

| Image encoder | Accuracy |
|---|---|
| Augmentation-aware | 23.25 |
| Augmentation-agnostic | **27.84** |

**Number of Augmentations**   We investigate the effect of the number of image views and text views that make up our multi-view batch for a given original image–text pair. As there exists a trade-off between the number of augmentations and the number of original image–text pairs in a batch, the number of image views and text views should be set to appropriate values for a balanced learning of intra-domain features and inter-domain features. As seen in Table F, using more text views is not helpful and actually hurts performance, which means that the benefits from increasing text views do not outweigh the losses due to decreased diversity in data samples, so we instead increase the number of image views. We have observed that the performance increases until the number of image views is 4, but the improvement was not that significant compared to the increased training time due to the decrease in the number of original pairs in a batch, so we decided to use 3 image views for faster training with acceptable performance. We leave it as a future work to find effective text augmentation methods in contrastive learning.

Table F: ImageNet-1k zero-shot accuracy with varying the number of image views and text views.

| # of image views | # of text views | # of original pairs | Accuracy |
|---|---|---|---|
| 1 | 1 | 192 | 21.80 |
| 2 | 1 | 128 | 25.54 |
| 2 | 2 | 96 | 24.60 |
| 3 | 1 | 96 | **27.67** |
| 3 | 2 | 72 | 24.57 |
| 4 | 1 | 72 | **28.25** |

# E  Misalignment Adjustments in UniCLIP

Several image–text pairs that are vulnerable to inter-domain misalignment due to augmentations and their similarity scores are presented in Tables G–J. In each table, there are two images and two captions, where the top image and the left caption are the original pair, the bottom image is an augmented image, and the right caption is a modified caption to pair with the augmented image.

For example in Table G, the image of yellow flowers is augmented to the image of orange flowers by a `ColorJitter` augmentation and as a consequence the corresponding caption on the left should be modified to the right one to correct the misalignment. Since CLIP and SLIP only employ `RandomResizedCrop` augmentation on image–text pairs, they do not experience severe inter-domain misalignment issues, resulting in higher similarity scores on the correct pairs. UniCLIP also produces correct results even if it has been trained with strong augmentations, which means the inter-domain misalignment problem is well addressed. In contrast, DeCLIP suffers from inter-domain misalignments and shows unpredictable results.

Interestingly, the projection head in UniCLIP can adjust inter-domain misalignments when information about the applied augmentation $\mathcal{A}$ is known to it via the augmentation encoder $f_A$. As in

Tables G–J, even for augmented images, UniCLIP can give the original captions higher similarity scores than the modified captions if augmentation information is provided by adjusting misalignment. For example in Table H, if the model knows that grayscale augmentation has been applied to the image and we let the model adjust the misalignment due to the augmentation, then it will try to guess what the color of the apples were if the image was an RGB image, which would be red with high probability in this case, thus putting a higher score on the original caption than the modified one.

Table G: Similarity scores between images and captions. Top row: original image, bottom row: jittered image.

| Image | Method | *"Flowers of **yellow** color."* | *"Flowers of **orange** color."* |
|---|---|---|---|
| | CLIP | **2.6907** | 2.6076 |
| | SLIP | **2.7534** | 2.7512 |
| | DeCLIP | 0.8485 | **0.8758** |
| | UniCLIP | **4.0810** | 2.6024 |
| | CLIP | 2.6213 | **2.8758** |
| | SLIP | 2.6075 | **2.8258** |
| | DeCLIP | 0.8412 | **0.8578** |
| | UniCLIP | 2.8236 | **3.8955** |
| | UniCLIP w/ $f_A(\mathcal{A})$ | **4.0754** | 2.7188 |

Table H: Similarity scores between images and captions. Top row: original image, bottom row: grayscale image.

| Image | Method | *"**Red** apples hanging from the tree."* | *"**Gray** apples hanging from the tree."* |
|---|---|---|---|
| | CLIP | **3.3721** | 2.7227 |
| | SLIP | **3.1031** | 2.4363 |
| | DeCLIP | **1.3140** | 1.1168 |
| | UniCLIP | **3.0319** | 2.6291 |
| | CLIP | 3.5971 | **4.4252** |
| | SLIP | 2.9678 | **3.2464** |
| | DeCLIP | **1.2837** | 1.1374 |
| | UniCLIP | 3.5288 | **3.9235** |
| | UniCLIP w/ $f_A(\mathcal{A})$ | **3.0319** | 2.6291 |

Table I: Similarity scores between images and captions. Top row: original image, bottom row: cropped image.

| Image | Method | *"A **tiny** chair."* | *"A **close-up** of a chair."* |
|---|---|---|---|
| | CLIP | **2.1118** | 1.9880 |
| | SLIP | **2.1535** | 1.8063 |
| | DeCLIP | **1.1831** | 1.0541 |
| | UniCLIP | **3.4089** | 2.9047 |
| | CLIP | 2.5919 | **3.2560** |
| | SLIP | 2.4766 | **2.6049** |
| | DeCLIP | **1.2042** | 1.0317 |
| | UniCLIP | 2.5598 | **2.7660** |
| | UniCLIP w/ $f_A(\mathcal{A})$ | **2.1417** | 1.9606 |

Table J: Similarity scores between images and captions. Top row: original image, bottom row: flipped image.

| Image | Method | *"From **left**, orange, mango, and apple."* | *"From **right**, orange, mango, and apple."* |
|---|---|---|---|
| | CLIP | **2.6101** | 2.5720 |
| | SLIP | **2.6285** | 2.5901 |
| | DeCLIP | 1.1800 | **1.2625** |
| | UniCLIP | **4.1929** | 3.9725 |
| | CLIP | 2.7728 | **2.8746** |
| | SLIP | 2.9061 | **2.9916** |
| | DeCLIP | 1.1615 | **1.2455** |
| | UniCLIP | 3.9946 | **4.0283** |
| | UniCLIP w/ $f_A(\mathcal{A})$ | **4.2233** | 4.0072 |

## F    Analysis

**Distribution of Similarities w.r.t Loss Functions**    Figure C shows distribution of similarities with respect to loss functions. MIL-NCE and SupCon losses show worse separation of positives–negatives compared to MP-NCE loss. In MIL-NCE (red), hard-positives are concentrated around a lower similarity region. SupCon loss (green) shows better separation of positives-negatives, but converges to decreased scores of easy-positives. MP-NCE (blue) shows the best separation of positives–negatives, as well as better convergence of easy-positives compared to SupCon loss. This result is consistent with the analysis in Section 2.2.

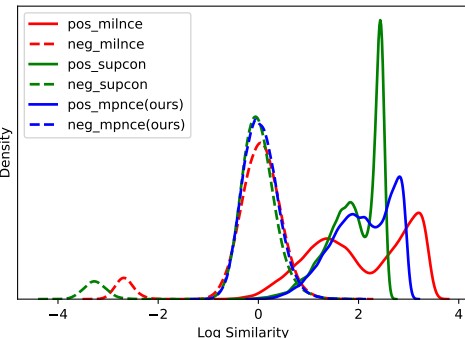

Figure C: Density plot of similarity scores with respect to loss functions.

**Distribution of Similarities w.r.t Domain Dependency of Similarity Measure**    Figure Da and Figure Db show distribution of similarities with respect to domain dependency of similarity measure. We can find that positives–negatives of image–image pairs separate better than positives–negatives of image–text pairs, which means the former pairs are easier to classify than the latter cases. With domain-dependent $\tau$ and $b$, those separations are more pronounced depending on the domain difficulties, resulting in better performance.

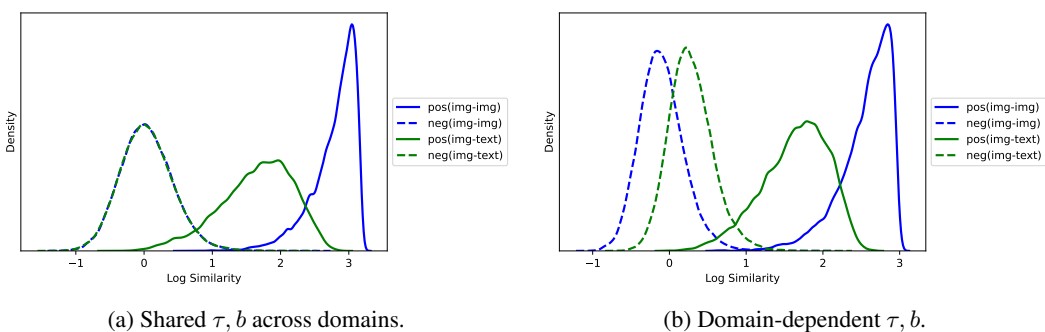

(a) Shared $\tau, b$ across domains.

(b) Domain-dependent $\tau, b$.

Figure D: Density plot of similarity scores with respect to domain dependency of similarity measure.

# G  Implementation Details

**Experimental Settings**  For the main experiments in Section 3.1, we used settings in Table K for UniCLIP training. For the baselines, we used learning rate and weight decay of (5e-4, 2e-1) for CLIP [32], (3e-3, 1e-1) for SLIP [24], and (1e-3, 1e-1) for DeCLIP [8], while the remaining hyperparameters are the same as our method. We used the implementation of CLIP and SLIP from https://github.com/facebookresearch/SLIP, and DeCLIP from https://github.com/Sense-GVT/DeCLIP. For a fair comparison, CLIP doubled the batch size to match memory usage. All models are trained with the automatic mixed precision in PyTorch [29]. Standard cropping and flipping augmentations [39] are used for linear probing, and RandAugment [7] is used for fine-tuning.

Table K: Training settings.

| | Pre-train | | Linear probing | Fine-tuning |
|---|---|---|---|---|
| Dataset → Config ↓ | YFCC15M | Open30M | 11 downstream | ImageNet |
| Base learning rate | 1e-3 | 1e-3 | 1e-1 | 5e-4 |
| Weight decay | 0.2 | 0.1 | 0 | 0.05 |
| Epoch | 50 | 32 | 90 | 100 |
| Linear warmup epoch | 2 | 1 | 0 | 5 |
| Learning rate schedule | Cosine decay | | - | - |
| Optimizer | AdamW | | SGD | AdamW |
| Optimizer momentum | 0.9, 0.98 | | 0.9 | 0.9, 0.999 |
| Total batch size | 4096 | | 128 | 256 |
| GPU | 16×A100 40GB | | 1×V100 16GB | 4×V100 16GB |

**Augmentation Configurations**  Following our ablation studies in Tables 4b and F, one weakly augmented image view, two strongly augmented image views, and one text is used to train our networks. Detailed image augmentation policies are described in Table L. For text augmentations, EDA [44] is applied only to CC3M since it has much more refined text data than other web-crawled noisy datasets like CC12M and YFCC15M. EDA is applied with a random replacement probability of 0.2 and a random deletion probability of 0.1.

Table L: Image augmentation configurations in PyTorch style.

| | Augmentation | Parameter | Value | Applying probability |
|---|---|---|---|---|
| Weak augmentation | RandomResizedCrop | size, scale, ratio | 224, [0.5, 1], [3/4, 4/3] | 1 |
| | ColorJitter | brightness, contrast, saturation, hue | 0.4, 0.4, 0.4, 0.1 | 0.8 |
| | GaussianBlur | kernel_size, sigma | 11, [0.1, 2] | 0.5 |
| Strong augmentation | RandomResizedCrop | size, scale, ratio | 224, [0.08, 1], [3/4, 4/3] | 1 |
| | ColorJitter | brightness, contrast, saturation, hue | 0.4, 0.4, 0.4, 0.1 | 0.8 |
| | GaussianBlur | kernel_size, sigma | 11, [0.1, 2] | 0.5 |
| | RandomHorizontalFlip | - | - | 0.5 |
| | RandomGrayscale | - | - | 0.2 |
| Strong augmentation (DeCLIP) | RandomResizedCrop | size, scale, ratio | 224, [0.2, 1], [3/4, 4/3] | 1 |
| | ColorJitter | brightness, contrast, saturation, hue | 0.4, 0.4, 0.4, 0.1 | 0.8 |
| | GaussianBlur | kernel_size, sigma | 11, [0.1, 2] | 0.5 |
| | RandomHorizontalFlip | - | - | 0.5 |
| | RandomGrayscale | - | - | 0.2 |

**Network Configurations**  Network configurations are summarized in Table M. The augmentation encoder is composed of 11-256-256-256 MLP with GELU activations. A residual block in the projection head is identical to the feedforward module in Transformers and ViTs. A linear layer follows 3 residual blocks in the projection head.

**Dataset Configurations**  Table N describes all dataset configurations used in our experiments.

Table M: Network configurations.

| | | Architecture | Input dimension | Output dimension | Transformer | | |
|---|---|---|---|---|---|---|---|
| | | | | | layers | width | heads |
| Image | Encoder $f_I$ | ViT-B/32 | 224×224 | - | 12 | 768 | 12 |
| | Augmentation encoder $f_A$ | 3-layer MLP | 11 | 256 | - | - | - |
| | Projection head $g_I$ | 3 ResBlocks | 1024 | 512 | - | - | - |
| Text | Encoder $f_T$ | Transformer | 77 | - | 12 | 512 | 8 |
| | Projection head $g_T$ | Linear | 512 | 512 | - | - | - |

Table N: Dataset configurations. Flickr30k and COCO Captions have 5 captions per image.

| | Dataset | # Classes | # Training | # Validation (# Test) |
|---|---|---|---|---|
| Pre-training | CC3M | - | 2,891,358 | - |
| | CC12M | - | 10,663,994 | - |
| | YFCC15M | - | 15,171,110 | - |
| | Open30M | - | 28,726,462 | - |
| Zero-shot classification & linear probing | Pets | 37 | 3,680 | 3,669 |
| | CIFAR-10 | 10 | 50,000 | 10,000 |
| | CIFAR-100 | 100 | 50,000 | 10,000 |
| | SUN397 | 397 | 19,850 | 19,850 |
| | Food-101 | 101 | 75,750 | 25,250 |
| | Flowers | 102 | 2,040 | 6,149 |
| | Cars | 196 | 8,144 | 8,041 |
| | Caltech-101 | 102 | 3,060 | 6,085 |
| | Aircraft | 100 | 6,667 | 3,333 |
| | DTD | 47 | 3,760 | 1,880 |
| | ImageNet | 1,000 | 1,281,167 | 50,000 |
| Image–text retrieval | Flickr30k | - | 31,784 | 1,000 (1,000) |
| | COCO Captions | - | 82,783 | 5,000 (5,000) |
| ImageNet variations | ImageNet-R | 200 | - | 30,000 |
| | ImageNet-Sketch | 1,000 | - | 50,000 |
| | ImageNetV2 | 1,000 | - | 30,000 |
| | ImageNet-A | 200 | - | 7,500 |