# OpenReview forum: "UniCLIP: Unified Framework for Contrastive Language-Image Pre-training"
_NeurIPS.cc/2022/Conference — NeurIPS 2022 Accept_

### Official Review · Reviewer_wcmt · 2022-07-11

**Rating:** 6
**Confidence:** 3
**Soundness:** 3 good
**Presentation:** 4 excellent
**Contribution:** 3 good

**Summary:**

In this work, the authors propose three components for contrastive learning across modalities which aim to remove discrepancies associated with learning representations with data augmentations, "hard" and "easy" positive pairs, and differences in similarity scoring across domains.

The first contribution is augmentation-aware feature embedding which concatenates an augmentation feature vector to image encodings before being fed through a projection head. This augmentation feature vector is used in order to constrain the image encoder to output representations that are "augmentation-agnostic" while allowing the projection head to make use of augmentation information.

The second contribution is a modified NCE loss (MP-NCE) which includes multiple positives and also uses a "strong positive" pair of an item with itself. Additionally, a rebalancing parameter is utilized to ensure that losses are equally weighted across all combinations of domains.

The third contribution is a domain dependent similarity measure which reweights similarly scores by introducing separate thresholds $b_{i,j}$  and temperature parameters $\tau_{i,j}$ for each pair of domains $i,j$.

**Questions:**

Please see Weakness 1 and 2

**Limitations:**

The authors have not adequately addressed the limitations and potential negative societal impact of their work and should do so in a final version of the paper.

**Strengths And Weaknesses:**

Strengths:

- The paper is well written and provides clear descriptions of each of the contributions as well as experimental results on well-studied datasets/benchmark models.

- Each of the contributions is well motivated by shortcomings in existing contrastive models/losses.

- Experimental results show clear improvements over recent works (CLIP, SLIP, DeCLIP)

- The experiment settings and hyperparameters are clearly outlined in Appendix Section C

Weaknesses:

- Exposition and presentation of each of the contributions ability to solve the problems they address is somewhat limited although proposed components are shown to be beneficial through ablation studies.. Some examples of the augmentation aware feature embedding is provided in Appendix Section E but it is unclear how the MP-NCE and the domain dependent similarity measure change the representations learned and their similarities. It would be helpful to understand how distribution of similarities changes with these mechanisms.

- Ethical considerations and limitations is not discusses, please consider including information on this material

---

> ### Author Response · Authors · 2022-08-02
> **Response to Reviewer wcmt**
>
>
> ### Distribution of similarities w.r.t loss functions
> Please refer to the `loss_ftn.png` file of the updated supplementary material for the histogram.
> MIL-NCE and SupCon losses show worse separation of positives-negatives compared to MP-NCE loss.
> In MIL-NCE (red), hard-positives are concentrated around lower similarity in the $x$-axis as expected in Line 166.
> SupCon loss (green) shows better separation of positives-negatives, but converges to decreased scores of easy-positives as noted in Line 180.
> MP-NCE (blue) shows best separation of positive-negatives, as well as better convergence of easy-positives compared to SupCon loss.
>
>
>
>
>
>
>
> ### Distribution of similarities w.r.t domain dependency of similarity measure
> Please refer to the `shared.png` and `domain_dependent.png` files of the updated supplementary material for the histograms.
> We can find that positives-negatives of image-image pairs separate better than positives-negatives of image-text pairs, which means the former pairs are easier to classify than the latter cases.
> With domain-dependent $\tau$ and $b$, those separations are more pronounced depending on the domain difficulties, resulting in better performance.
>
> ### Limitations
>
> Works on learning vision-language modalities can be grouped into encoder-decoder architecture methods and dual-encoder architecture methods. Captioning and VQA tasks can be conducted in encoder-decoder methods, however, our proposed method falls under the dual-encoder category. Works in this category aim for zero-shot classification and retrieval tasks through cross-modal alignment training.
> More recent literature [1] involves joining the above two categories. As a future work for extension to image captioning and VQA, we believe that one can employ our method in CoCa [1] by substituting the contrastive loss to further improve performance.
>
> In this paper, image-text datasets were used to validate our method since vision and language are among the most actively studied fields in deep learning.
> Although we have experimented with vision-language multimodal datasets only, the proposed UniCLIP framework can be easily extended to other types of multimodal datasets because it is designed in a modality-agnostic way except for the augmentation encoding part.
> All modality-specific knowledge required to apply UniCLIP to different types of modality is to describe each modality-specific augmentation as a real vector, as in Lines 97-109, which is quite straightforward.
> For examples, with audio datasets, if you change the pitch of an audio by $p\in [-P, P]$ semitones randomly then you can describe this augmentation as $(p)$, and if you randomly reverse an audio then you can describe it as $(0)$/$(1)$ flag vector, etc.
> It would be great if we could see how the UniCLIP framework works with various types of multimodal datasets from machine learning community.
>
> [1] Yu, Jiahui, et al. "Coca: Contrastive captioners are image-text foundation models." arXiv preprint arXiv:2205.01917 (2022).
>
> ### Potential negative societal impacts
> We do not anticipate potential negative societal impacts.
> Our method enables strong data augmentation methods and makes use of all possible pairs in batches, so it is actually more data-efficient than other related methods.

---

> > ### Comment · Reviewer_wcmt · 2022-08-08
> > **Response to Rebuttal**
> >
> > Thanks for providing detailed answers to my questions, limitations provided are very detailed and I believe the similarities will be useful for readers and the authors should consider adding them to the appendix. Also, I appreciate that the authors have conducted finetuned retrieval tasks during the discussion period.
> >
> > The paper is well written, experiments are easy to follow, and the proposed method provides performance improvements, however the contributions are somewhat incremental as pointed out by other reviewers. I have kept my score at 6.

---

### Official Review · Reviewer_xbYy · 2022-07-11

**Rating:** 6
**Confidence:** 3
**Soundness:** 3 good
**Presentation:** 4 excellent
**Contribution:** 3 good

**Summary:**

The paper proposes learning a CLIP model in a single representation space to overcome the limitations of previous work on independent learning of each self-supervised objectives. The model, which they call UniCLIP, integrates the contrastive loss of both inter-domain pairs and intra-domain pairs into a single universal space. The authors utilize three methods to address the discrepancy between different domains: augmentatio-aware feature embedding, an MP-NCE loss, and a domain-dependent similarity measure.



**Questions:**

1) A detailed discussion of limitations is needed.
2) Need experimental validation of the claim that UniCLIP can be used in various types of multimodal datasets. How can your claim be justified with such experiments?


**Limitations:**

The authors do not properly address the limitations of their work. I encourage the author to directly address that in their rebuttal.

**Strengths And Weaknesses:**

Strengths:
(1) The paper is well-written.
(2) Experimental settings are described in detail, aiding reproducibility.
(3) The proposed method leads to significant improvements on zero-shot, linear probing and retrieval test.
(4) Strong ablation studies.

Weaknesses:
(1) This work builds incrementally on previous works.
(2) Experiments on a wider range of tasks are encouraged, e.g., image captioning and visual question answering, as the method claims to learn representations in a unified space, improvements on visual-language tasks are expected.
(3) Lack of proper discussion of limitations.
(4) Some of the claims are not backed by experiments, for instance, the claim that UniCLIP can be used in various types of multimodal datasets.
(5) The paper has several typos (e.g., "datsets")

---

> ### Author Response · Authors · 2022-08-02
> **Response to Reviewer xbYy**
>
> ### Limitations
>
> Works on learning vision-language modalities can be grouped into encoder-decoder architecture methods and dual-encoder architecture methods. Captioning and VQA tasks can be conducted in encoder-decoder methods, however, our proposed method falls under the dual-encoder category. Works in this category aim for zero-shot classification and retrieval tasks through cross-modal alignment training.
> More recent literature [1] involves joining the above two categories. As a future work for extension to image captioning and VQA, we believe that one can employ our method in CoCa [1] by substituting the contrastive loss to further improve performance.
>
> In this paper, image-text datasets were used to validate our method since vision and language are among the most actively studied fields in deep learning.
> Although we have experimented with vision-language multimodal datasets only, the proposed UniCLIP framework can be easily extended to other types of multimodal datasets because it is designed in a modality-agnostic way except for the augmentation encoding part.
> All modality-specific knowledge required to apply UniCLIP to different types of modality is to describe each modality-specific augmentation as a real vector, as in Lines 97-109, which is quite straightforward.
> For examples, with audio datasets, if you change the pitch of an audio by $p\in [-P, P]$ semitones randomly then you can describe this augmentation as $(p)$, and if you randomly reverse an audio then you can describe it as $(0)$/$(1)$ flag vector, etc.
> It would be great if we could see how the UniCLIP framework works with various types of multimodal datasets from machine learning community.
>
> We would like to thank you for pointing out some limitations of this study and for helping to complete the paper with higher quality.
>
> [1] Yu, Jiahui, et al. "Coca: Contrastive captioners are image-text foundation models." arXiv preprint arXiv:2205.01917 (2022).
>
>
> ### Typos
> Fixed the typo.

---

> ### Comment · Reviewer_xbYy · 2022-08-07
> **Thank you for your response**
>
> Thanks for the response about the limitations. Do you have any response to the following that as mentioned in the original review:
>
>
> 1-Experiments on a wider range of tasks are encouraged, e.g., image captioning and visual question answering, as the method claims to learn representations in a unified space, improvements on visual-language tasks are expected.
>
> 2- Need experimental validation of the claim that UniCLIP can be used in various types of multimodal datasets. How can your claim be justified with such experiments?

---

> > ### Author Response · Authors · 2022-08-08
> > **Response to Reviewer xbYy**
> >
> > ### 1
> > As we have mentioned in the limitations, works on learning vision-language representations can be grouped into encoder-decoder architecture methods and dual-encoder architecture methods [1], *regardless* of whether the latent space is unified or separated.
> > [2, 3, 4, 5] and UniCLIP belong to dual-encoder methods, in which image representations and text representations are learned through image encoders and text encoders respectively, whereas encoder-decoder methods include extra decoders to transform learned representations into targets such as texts for image captioning and VQA.
> > Therefore extending a dual-encoder model to an encoder-decoder model requires designing a new decoder architecture and also training the decoder from scratch for the new downstream tasks like VQA, which is not that straightforward and is another line of research as you can see in [1, 6].
> > Thus zero-shot or fine-tuned image classification tasks and image-text retrieval tasks are widely used to evaluate dual-encoder models as single-modal and multi-modal downstream tasks respectively, while image captioning and VQA tasks are used for encoder-decoder models.
> > We can directly employ our method in encoder-decoder methods like [1] by substituting the contrastive loss by UniCLIP with excessive experiments, but it is out of scope as we are proposing a dual-encoder method, *not* an encoder-decoder method, and the plentiful experiments in our paper already demonstrate other dual-encoder methods and ours sufficiently, so it should rather be one of the future works to apply our method to encoder-decoder methods for image captioning and VQA.
> >
> > [1] Yu, Jiahui, et al. "Coca: Contrastive captioners are image-text foundation models." arXiv preprint arXiv:2205.01917 (2022).\
> > [2] Radford, Alec, et al. "Learning transferable visual models from natural language supervision." International Conference on Machine Learning. PMLR, 2021.\
> > [3] Jia, Chao, et al. "Scaling up visual and vision-language representation learning with noisy text supervision." International Conference on Machine Learning. PMLR, 2021.\
> > [4] Mu, Norman, et al. "Slip: Self-supervision meets language-image pre-training." arXiv preprint arXiv:2112.12750 (2021).\
> > [5] Li, Yangguang, et al. "Supervision exists everywhere: A data efficient contrastive language-image pre-training paradigm." arXiv preprint arXiv:2110.05208 (2021).\
> > [6] Alayrac, Jean-Baptiste, et al. "Flamingo: a visual language model for few-shot learning." arXiv preprint arXiv:2204.14198 (2022).
> >
> > ### 2
> > As we have mentioned in the limitations, the proposed UniCLIP framework can be easily extended to other types of multimodal datasets because it is designed in a modality-agnostic way in most parts.
> > We have fully demonstrated our method on image-text datasets which are the most actively researched multimodal datasets, and the method descriptions and all the experiments already exceed the page limit of a conference paper, so we leave it for future work.

---

> > > ### Comment · Reviewer_xbYy · 2022-08-08
> > > **Thank you**
> > >
> > > Thanks for the clarification. I have changed my score from 5 to 6.

---

### Official Review · Reviewer_q238 · 2022-07-11

**Rating:** 5
**Confidence:** 3
**Soundness:** 2 fair
**Presentation:** 3 good
**Contribution:** 2 fair

**Summary:**

This paper propose a framework called UniCLIP to integrate the contrastive loss of both inter-domain pairs and intra-domain pairs into a single universal space for Vision-Language Contrastive Pre-training. This work is more like a fusion of three optimizations than other work: augmentation12 aware feature embedding, (2) MP-NCE loss, and (3) domain dependent similarity measure.
The results of UniCLIP performs better than previous work.

**Questions:**

N/A

**Ethics Review Area:**

["I don’t know"]

**Strengths And Weaknesses:**

Strengths:
Good performance and simple
This paper is will written and organized.
Weaknesses:
Lack of innovation, this paper is more like a combinatorial work, (2) MP-NCE loss, and (3) domain dependent similarity measure These two are not original to this paper, and (1) data augmentation is also a very common technique.

---

> ### Author Response · Authors · 2022-08-02
> **Response to Reviewer q238**
>
> ### Novelty
> Even though data augmentation is a very common technique, the augmentations used in prior vision-language pre-training such as CLIP and SLIP had been limited to a set of weak transformations due to the misalignment problem described in Section 1.
> Our design of augmentation-agnostic encoder and augmentation-aware head is a novel framework in the large scale vision-language pre-training literature; this enables it to employ strong augmentation transformations without suffering from the misalignment problem, as opposed to the prior works such as DeCLIP.
> As more positive pairs can be effectively used with stronger augmentations, the training discrepancy between positive pairs and between domains - which had been overlooked in other multi-modal contrastive learning frameworks - become significantly pronounced. This has led to us to overhaul existing multi-positive contrastive loss functions and construct MP-NCE loss and domain dependent similarity measure.

---

### Official Review · Reviewer_rSQy · 2022-07-11

**Rating:** 5
**Confidence:** 3
**Soundness:** 3 good
**Presentation:** 3 good
**Contribution:** 2 fair

**Summary:**

The paper proposes UniCLIP (Unified Framework for Contrastive Language–Image Pre-training) to unify inter- and intra-modality contrastive learning under a joint pretraining objective. The main difference of UniCLIP from prior work is that UniCLIP considers all inter- and intra-modal positives/negatives in the contrastive loss. Specifically, as an example, when computing loss for a text-image pair, image-image and text-text negative pairs are also considered. A dedicated multi-positive NCE loss is used to highlight the gradient of hard positives, which is in theory and empirically better than other options such as MIL-NCE loss and Supervised Contrastive Loss. Another special design is having a separate augmentation encoder inserting augmentation information in between the image encoder and image projection head so that the image encoder is agnostic to data augmentations while augmentation’s change to image semantics is also taken into consideration in the projected embedding space.


**Questions:**

* For the baselines such as SLIP and DeCLIP, did you implement your own versions and train your own models? Sorry if I missed this detail.

* Since the pretraining dataset is different from the original configuration of DeCLIP, the numbers reported in the original paper are not directly comparable to this paper. I find it a bit puzzling that in some recent reports (https://arxiv.org/pdf/2203.05796.pdf), DeCLIP outperforms SLIP by large margins while in this paper, they perform similarly. Any comments on this?


**Limitations:**

I do not see any major issues.

**Strengths And Weaknesses:**


Strenghts

+ The paper has done a comprehensive and rigorous discussion of available contrastive loss functions. The explanations are clear in pointing out each specific configuration’s impact in the context of image-text contrastive learning.

+ Zero-shot classification performance seems to be strong and improvement is consistent across most of the 11 tasks.


Weaknesses

- Why not compare finetuning results on image-text retrieval? Zero-shot performance is not entirely reliable for comparing cross-modal retrieval capability. In my experience, sometimes finetuning even with few-shot examples can result in a huge difference.

- I find the idea of learning discriminative features by leveraging positives/negatives both within and across modalities not drastically different from prior works. Beyond vision-language pretraining, the choice of positives/negatives, and what to include/exclude is comprehensively explored in metric learning or earlier data mining literature. It is good to know that such design works well for vision-language pretraining, but it is not an exceptionally novel contribution.

---

> ### Author Response · Authors · 2022-08-02
> **Response to Reviewer rSQy**
>
> ### Fine-tuning results on image-text retrieval
>
> We fine-tuned YFCC15M pre-trained models on Flickr30k and COCO Captions for 10 epochs and report the results in the `finetuned_retrieval.png` file of the updated supplementary material.
> Our method consistently outperforms on fine-tuned image-text retrieval benchmarks.
>
> ### Metric learning and earlier data mining literature
>
> Thank you for pointing out link to metric learning and data mining literature. While we agree that vision-language pre-training and self-supervised learning (SSL) are closely related to metric learning, we would like to highlight some notable deviations of our work from traditional cross-modal metric learning.
>
> Traditional metric learning concerns learning cross-modal embedding from inter-modal example pairs. Whereas, recent SSL methods learn transferable network initialization by learning from synthetic intra-modal example pairs [5, 6]. Our work is directed towards integrating cross-modal metric learning and SSL, which opens an unexplored search space for pair selection problem. Note that earlier works in metric learning have addressed inter-modal learning but not intra-modal learning [1, 2, 3, 4]. Some subsequent works jointly perform intra-modal and inter-modal learning, but the augmented examples were neglected in inter-modal learning signals [7, 8, 9]. Our work identifies the misalignment problem existing in the inter-modal pairs between augmented samples. Moreover, we proposed an extension of contrastive loss that leverages learning signals from all possible inter-modal pair combinations that fully incorporates the augmented examples. This allows UniCLIP to achieve state-of-the-art in vision-language embeddings and pre-trained initialization.
>
> [1] Wei, Jiwei, et al. "Universal weighting metric learning for cross-modal matching." Proceedings of the IEEE/CVF conference on computer vision and pattern recognition. 2020.\
> [2] Mei, Xinhao, et al. "On Metric Learning for Audio-Text Cross-Modal Retrieval." arXiv preprint arXiv:2203.15537 (2022).\
> [3] Wang, Jian, et al. "Image-text cross-modal retrieval via modality-specific feature learning." Proceedings of the 5th ACM on International Conference on Multimedia Retrieval. 2015.\
> [4] Karpathy, Andrej, Armand Joulin, and Li F. Fei-Fei. "Deep fragment embeddings for bidirectional image sentence mapping." Advances in neural information processing systems 27 (2014).\
> [5] Chen, Ting, et al. "A simple framework for contrastive learning of visual representations." International conference on machine learning. PMLR, 2020.\
> [6] Gao, Tianyu, et al. “SimCSE: Simple Contrastive Learning of Sentence Embeddings.” Empirical Methods in Natural Language Processing (EMNLP), 2021.\
> [7] Mikriukov, Georgii, et al. “Unsupervised Contrastive Hashing for Cross-Modal Retrieval in Remote Sensing.” ICASSP 2022 - 2022 IEEE International Conference on Acoustics, Speech and Signal Processing (ICASSP), 2022, pp. 4463–67, doi:10.1109/ICASSP43922.2022.9746251.\
> [8] Mu, Norman, et al. "Slip: Self-supervision meets language-image pre-training." arXiv preprint arXiv:2112.12750 (2021).\
> [9] Li, Yangguang, et al. “Supervision Exists Everywhere: A Data Efficient Contrastive Language-Image Pre-Training Paradigm.” International Conference on Learning Representations, 2022, https://openreview.net/forum?id=zq1iJkNk3uN.
>
>
>
>
> ### SLIP and DeCLIP implementations
> For SLIP and DeCLIP, we used the implementations in their official code repositories [1, 2], where we have made minimal modifications to make the codes executable in our machines.
> In [3], all methods are trained with the same hyperparameters, which can output suboptimal results for some methods.
> In contrast, we have followed the original hyperparameter settings suggested in each paper to reproduce optimal performances for each method.
> As a result, the margins reported by [3] appear to be slightly overstated than expected.
>
> [1] https://github.com/facebookresearch/SLIP \
> [2] https://github.com/Sense-GVT/DeCLIP \
> [3] Cui, Yufeng, et al. "Democratizing Contrastive Language-Image Pre-training: A CLIP Benchmark of Data, Model, and Supervision." arXiv preprint arXiv:2203.05796 (2022).

---

### Meta-Review · Area_Chair_P3XW · 2022-08-26

**Recommendation:** Accept
**Confidence:** Certain

**Metareview:**

This paper proposed a framework for image-language pretraining which include three techniques.  The techniques are well motivated and developed to achieve better performance on zero-shot, linear probing and retrieval test.  Ablation studies are provided to show the effectiveness of the three techniques.  The writing of the paper is clear.  The authors appropriately answered most of the questions of the reviewers.

**Award:**

No

---

### Decision · Program_Chairs · 2022-09-14

Accept